# *Lacticaseibacillus paracsei* HY7207 Alleviates Hepatic Steatosis, Inflammation, and Liver Fibrosis in Mice with Non-Alcoholic Fatty Liver Disease

**DOI:** 10.3390/ijms25189870

**Published:** 2024-09-12

**Authors:** Hyeon-Ji Kim, Hye-Jin Jeon, Dong-Gun Kim, Joo-Yun Kim, Jae-Jung Shim, Jae-Hwan Lee

**Affiliations:** R&BD Center, hy Co., Ltd., 22, Giheungdanji-ro 24beon-gil, Giheung-gu, Yongin-si 17086, Republic of Korea; skyatk94@gmail.com (H.-J.K.); 10003012@hy.co.kr (H.-J.J.); kimdg@hy.co.kr (D.-G.K.); jjshim@hy.co.kr (J.-J.S.); jaehwan@hy.co.kr (J.-H.L.)

**Keywords:** non-alcoholic fatty liver disease, probiotics, *Lacticaseibacillus paracasei*, steatosis, lipogenesis, apoptosis, NASH, NAFLD

## Abstract

Non-alcoholic fatty acid disease (NAFLD) is caused by a build-up of fat in the liver, inducing local inflammation and fibrosis. We evaluated the effects of probiotic lactic acid-generating bacteria (LAB) derived from a traditional fermented beverage in a mouse model of NAFLD. The LAB isolated from this traditional Korean beverage were screened using the human hepatic cell line HepG2, and *Lactocaseibacillus paracasei* HY7207 (HY7207), which was the most effective inhibitor of fat accumulation, was selected for further study. HY7207 showed stable productivity in industrial-scale culture. Whole-genome sequencing of HY7207 revealed that the genome was 2.88 Mbp long, with 46.43% GC contents and 2778 predicted protein-coding DNA sequences (CDSs). HY7207 reduced the expression of lipogenesis and hepatic apoptosis-related genes in HepG2 cells treated with palmitic acid. Furthermore, the administration of 10^9^ CFU/kg/day of HY7207 for 8 weeks to mice fed an NAFLD-inducing diet improved their physiologic and serum biochemical parameters and ameliorated their hepatic steatosis. In addition, HY7207 reduced the hepatic expression of genes important for lipogenesis (*Srebp1c*, *Fasn*, *C*/*ebpa*, *Pparg*, and *Acaca*), inflammation (*Tnf*, *Il1b*, and *Ccl2*), and fibrosis (*Col1a1*, *Tgfb1*, and *Timp1*). Finally, HY7207 affected the expression of the apoptosis-related genes *Bax* (encoding Bcl2 associated X, an apoptosis regulator) and *Bcl2* (encoding B-cell lymphoma protein 2) in the liver. These data suggest that HY7207 consumption ameliorates NAFLD in mice through effects on liver steatosis, inflammation, fibrosis, and hepatic apoptosis. Thus, *L. paracasei* HY7207 may be suitable for use as a functional food supplement for patients with NAFLD.

## 1. Introduction

Non-alcoholic fatty liver disease (NAFLD) is a disease in which excessive fat accumulates in the liver in the absence of excessive alcohol intake or viral hepatitis. NAFLD is characterized by triglyceride accumulation in ≥5% of hepatocytes and is a leading cause of chronic liver disorders. It comprises steatosis, non-alcoholic steatohepatitis (NASH), liver fibrosis, and cirrhosis [1,2,3]. The global prevalence of NAFLD is estimated to be approximately 32%, and it is more prevalent in men than in women (33% in men and 20% in women) [4,5]. Treatment strategies for NAFLD and NASH are being researched worldwide, but appropriate treatments are currently lacking [6,7]. Accordingly, it is necessary to study the key mechanisms of NAFLD to help develop superior methods of diagnosing and treating the constituent diseases.

De novo lipogenesis (DNL) increases the transport and accumulation of fatty acids in the liver [8]. The lipogenesis pathway is principally regulated by sterol regulatory element-binding protein-1 (SREBP-1C) and peroxisome proliferator-activated receptor γ (PPARγ) [9,10]. SREBP-1c induces the synthesis of triglycerides and cholesterol by promoting the activities of acetyl-CoA carboxylase (ACC) and fatty acid synthase (FAS), resulting in fat accumulation in the liver [11,12]. Moreover, greater triglyceride synthesis in the liver increases low-density lipoprotein (LDL) release from the liver into the blood, resulting in dyslipidemia [13,14]. Therefore, to prevent the progression of NAFLD, it is important to diagnose the disease early and treat the hyperlipidemia. In addition, hepatocyte apoptosis, one of key components of NAFLD, is considered to be the primary cause of liver inflammation and fibrosis [15,16,17]. This is mediated through the upregulation of Bcl-2 family proteins and caspase-induced hepatocyte apoptosis [18]. Furthermore, the progression of NAFLD to NASH induces an increase in the production of pro-inflammatory substances, such as tumor necrosis factor (TNF) and CC-chemokine ligand (CCL-2) [19,20]. In addition, liver inflammation is a major precursor of fibrosis, which is largely mediated by transforming growth factor-β (TGF-β) [21].

Probiotics were defined as “Live microorganisms that when administered in adequate amounts confer a health benefit on the host” by the World Health Organization (WHO) in 2001 [22]. Most probiotics are known to have beneficial effects on the health of the host, as well as to have beneficial effects on the gastrointestinal tract and intestinal microbial balance [23,24]. *Lacticaseibacillus paracasei* (*L. paracasei*) is a probiotic strain that has been identified in dairy products and fermented foods and has been used safely in the food industry for a long time. *L. paracasei* has various health-promoting effects, such as improving intestinal health, reducing inflammation, acting as an antioxidant, and ameliorating allergic respiratory disease and inflammatory bowel disease (IBD) [25,26,27,28]. In addition, recent studies have shown that *L. paracasei* ameliorates obesity and NAFLD by suppressing NAFLD-related inflammation and reducing liver fat accumulation in high-fat-diet-fed rodents [29,30]. This suggests that probiotics may be one of the most effective treatments for these conditions.

The purpose of this study was to determine whether the new potential probiotic strain *Lacticaseibacillus paracasei* HY7207 (HY7207), isolated from Korean fermented beverages, ameliorates NAFLD. Fifteen strains of *L. paracasei* were screened for their effects to inhibit lipid accumulation in palmitic acid (PA)-treated HepG2 cells, and HY7202 was found to be the most effective. Subsequently, the industrial growth curve of this strain was evaluated for use in the food industry, and whole-genome sequencing was performed. We also evaluated the anti-lipogenic and anti-apoptotic effects of HY7207 in vitro and in mice fed an NAFLD-inducing diet. Finally, we performed a histopathologic analysis to evaluate the amelioration of the NAFLD activity score (NAS) in *L. paracasei* HY7207-treated mice with NAFLD.

## 2. Results

### 2.1. Screening of LAB Strains Isolated from Traditional Fermented Beverages

We isolated a total of 15 lactic acid-generating bacterial (LAB) strains from a Korean traditional fermented beverage (six strains of *Lacticaseibacillus paracasei* and ten strains of *Lactiplantibacillus plantarum*). The isolated LAB strains were then evaluated with respect to their effects on lipid accumulation in PA-treated HepG2 cells. As shown in Figure 1A, lipid accumulation in the PA-treated cells (137.4% ± 10.57%) was significantly more substantial than in the control cells (*p* < 0.01). The #5 strain (114.63% ± 4.22%) was the only one to significantly reduce lipid accumulation in these HepG2 cells (*p* < 0.05). Therefore, we selected *L. paracasei* #5 for further study and renamed this strain *Lacticaseibacillus paracasei* HY7207 (HY7207).

### 2.2. Growth Curve of HY7207 in Industrial Culture

To determine whether HY7207 is suitable for commercial use, the strain was cultured using an industrial fermenter, then we measured bacterial growth in terms of colony-forming units (CFUs)/mL. In addition, the growth curve for HY7207 reached the stationary phase after 21 h in a large fermentation volume (1200 L).

### 2.3. Whole-Genome Sequencing of HY7207

Whole-genome sequencing of *L. paracasei* HY7207 showed that it consisted of a circular chromosome of 2,877,365 bp, with a G+C content of 46.43% (Table 1). This genome contained a predicted 2778 CDSs, 15 rRNA-encoding genes, and 58 tRNA-encoding genes. The predicted CDSs were divided into clusters of orthologous gene (COG) functional categories as follows: T (56), U (19), V (74), C (98), D (26), E (190), F (79), G (264), H (42), I (49), J (148), K (183), L (190), M (126), N (1), O (59), P (115), Q (17), and S (694). The A, B, R, W, X, Y, and Z categories were not classified. Furthermore, the sequencing of HY7207 confirmed no antibiotic resistance genes. The HY7207 genome map is shown in Figure 1C.

### 2.4. Effects of HY7207 on the Expression of Genes Involved in Lipogenesis and Apoptosis by PA-Treated HepG2 Cells

We determined the effects of HY7207 on the expression of genes related to lipogenesis in PA-treated HepG2 cells. For in vitro experiments, 10^6^ and 10^7^ CFU/mL of HY7207 were added to the lipid accumulation-induced hepatic cell models. As shown in Figure 2A–C, the expression of *SREBP-1c* (3.78-fold, *p* < 0.001), *FASN* (1.87-fold, *p* < 0.001), and *C*/*EBPα* (1.23-fold, *p* < 0.05) in PA-treated cells was significantly higher than that of the untreated cells. However, the expression of *SREBP-1c* and *FASN* in the HY7207-treated group was significantly lower than that of the PA-treated group (*p* < 0.001). The mRNA expression of *C*/*EBPα* tended to be lower in the 10^6^ CFU/mL HY7207-treated group, but this difference was not significant. However, 10^7^ CFU/mL HY7207 treatment significantly reduced the mRNA expression of *C*/*EBPα* vs. the PA-treated cells. Overall, HY7207 treatment reduced the expression of lipogenesis-related genes in a dose-dependent manner.

Next, we measured the expression of the apoptosis-related genes *BAX* and *BCL-2* (Figure 2D,E). The expression of *BAX* in the PA-treated group was significantly higher than that in the untreated group (*p* < 0.001), but that in the HY7207-treated group was significantly lower than that of the PA-induced group. By contrast, the mRNA expression of *BCL-2* did not differ between the groups. In Figure 2F, the *BAX*/*BCL-2* ratio was significantly higher in cells treated with PA than in untreated cells (*p* < 0.01), but HY7207 treatment reduced this. As illustrated in Figure 2G,H, the mRNA expression of *CASP3* and *CASP9* in the PA-treated group was significantly elevated compared to that untreated group (*p* < 0.001). Treatment of HY7207 showed a dose-dependent manner reduction effect. These results indicate that HY7207 reduces the expression of lipogenesis-related genes and ameliorates the abnormal expression of apoptosis-related genes.

### 2.5. Effects of HY7207 on Physiologic Parameters in Mice Fed an NAFLD-Inducing Diet

To investigate the effects of HY7207 on NAFLD, we fed mice with an NAFLD-inducing diet and used metformin (MFM) as a positive control, because it is known to reduce hepatic fat synthesis and accumulation and to suppress glucose production [31,32]. In animal experiments, the MFM and HY7207-treated group orally ingested 250 mg/kg/day of metformin and 10^9^ CFU/kg/day of *L. paracasei* HY7207, respectively. We measured the changes in body mass and food intake during the animal experiment (Figure 3 and Appendix A).

As shown in Figure 3B, the body mass gain of the mice in the NAFLD-inducing diet (ND)-fed group was significantly higher (18.43 ± 3.44 g, *p* < 0.001) than that of the control group (10.13 ± 2.13 g). Both MFM and HY7207 significantly reduced this body mass gain, to 9.59 ± 1.94 g (*p* < 0.001) and 13.82 ± 1.88 g (*p* < 0.01), respectively, vs. the ND group.

As illustrated in Figure 3C, the food efficiency ratio (FER) for the ND group was also significantly higher (77.19% ± 10.81%, *p* < 0.001) than that of the control group (CON, 41.38% ± 6.47%). However, the FERs for the positive control and HY7207-treated groups were significantly lower (46.17% ± 8.40% (*p* < 0.001) and 59.66% ± 6.86% (*p* < 0.01), respectively) than that of the ND group.

Figure 3D,F shows the liver and epididymal fat masses of the mouse groups. The ND group had significantly higher masses than the CON group (*p* < 0.001), but these were significantly lower in the MFM and HY7207 groups than in the ND group.

In addition, the liver/body mass ratio (L/B ratio) of the ND group was significantly higher (6.78% ± 1.23%, *p* < 0.001) than that of the control group (4.07% ± 0.14%). The L/B ratios of the MFM and HY7207 groups were significantly lower (5.37% ± 0.54% (*p* < 0.05) and 5.50% ± 0.70% (*p* < 0.05)) than that of the ND group (Figure 3E). The epididymal fat/body mass ratio (E/B ratio) was significantly increased (6.47% ± 0.34%, *p* < 0.001) by the ND, and this was reduced to 5.21% ± 0.66% (*p* < 0.001) and 5.78% ± 0.32% (*p* < 0.05) by MFM and HY7207 treatment, respectively (Figure 3G).

### 2.6. Effects of HY7207 on Blood Biochemistry

Next, we measured levels of liver function markers (AST, ALT, ALP, and γ-GPT) in the serum of the mice (Figure 4A–D). The activities in the ND group were significantly higher than those in the CON group (AST, ALT, ALP, *p* < 0.001; γ-GPT, *p* < 0.001). MFM treatment restored these activities to levels similar to those of the control group, and HY7207 significantly reduced these activities vs. those of the ND group.

As shown in Figure 4E–G, the serum TG, T-CHO, and LDL-C concentrations were significantly higher in the ND group than in the CON group, and MFM and HY7207 significantly reduced the concentrations of all three vs. the ND group. HY7207 treatment also tended to reduce these concentrations, but there were no significant differences. In addition, the HDL-C concentration was slightly higher in the ND, MFM, and HY7207 groups than in the CON group, but the differences were not significant (Figure 4H). Finally, the GLU concentration of the ND group was slightly higher than that of the control group (Figure 4I), and MFM significantly reduced this vs. the ND group (*p* < 0.01).

### 2.7. Effects of HY7207 on the Histology of the Liver

To analyze the effect of HY7207 on hepatic morphology and histology, we collected the liver from each mouse. The morphology of the livers is shown in Figure 5A. We then examined sections of each liver after hematoxylin and eosin (H&E) staining, and as shown in Figure 5B, the ND group showed hepatic steatosis, but MFM and HY7207 treatment reduced this. Figure 5C shows that the high hepatic steatosis score of the ND group was ameliorated by MFM and HY7207. In particular, the administration of HY7207 significantly reduced the degree of steatosis induced by the NALFD diet (*p* < 0.05). The score of lobular inflammation and hepatocyte ballooning in the MFM and HY7207 ingestion group was also reduced compared to the ND group (Appendix A). In addition, as illustrated in Figure 5D, the NAS grade of the ND group was significantly higher than that of the CON group (*p* < 0.001), but that of the HY7207 group was significantly lower than that of the ND group (*p* < 0.05). Thus, HY7207 ameliorates NAFLD in mice.

### 2.8. Effects of HY7207 on the Hepatic Gene Expression of Mice with NAFLD

We next evaluated the effects of HY7207 on the mRNA expression of genes related to lipid synthesis, inflammation, and fibrosis in NAFLD-inducing diet-fed mice. First, we measured the expression of the lipogenesis-related genes *Srebp-1c*, *Fasn*, *C*/*ebpα*, *Pparγ*, and *Acaca*. As shown in Figure 6A, the expression of *Srebp-1c* (2.77-fold, *p* < 0.001), *Fasn* (2.77-fold, *p* < 0.001), *Pparγ* (4.49-fold, *p* < 0.001), and *Acaca* (1.33-fold, *p* < 0.05) was significantly higher in the ND group than in the control group. The expression of *C*/*ebpα* was also high in the ND group (1.24-fold), but the difference was not significant. However, the mRNA expression of *Srebp-1c*, *Fasn*, *Pparγ*, *Acaca* (*p* < 0.001), and *C*/*ebpα* (*p* < 0.05) was significantly lower in the MFM group. HY7207 also significantly reduced the expression of lipogenesis-related genes vs. the ND group (*Srebp-1c*, *Fasn*, *Acaca*, *p* < 0.001; *C*/*ebpα*, *Pparγ*, *p* < 0.01). In particular, HY7207 reduced the mRNA expression of *Srebp-1c* (0.68-fold), *C*/*ebpα* (0.78-fold), and *Acaca* (0.67-fold) to levels similar to those achieved by MFM (*Srebp-1c*, 0.96-fold; *C*/*ebpα*, 0.83-fold; *Acaca*, 0.63-fold).

In addition, we measured the expression of inflammation-related genes, such as those encoding pro-inflammatory cytokines (*Tnf* and *Il-1β*), and a chemokine (*Ccl2*) (Figure 6B). The ND group showed significantly higher expression of these genes than the control group, and these differences were significantly different with respect to *Tnf* and *Ccl2* (*p* < 0.001). The positive control, MFM, significantly altered the expression of *Tnf*, *Il-1β*, and *Ccl2* by 1.61-fold (*p* < 0.01), 0.53-fold (*p* < 0.05), and 1.38-fold (*p* < 0.001), respectively, vs. the ND group (*Tnf*, 3.35-fold; *Il-1β*, 1.19-fold; and *Ccl2*, 4.59-fold). In addition, the HY7207 group had significantly lower expression of the three genes (1.03-fold (*p* < 0.001), 0.47-fold (*p* < 0.01), and 1.54-fold (*p* < 0.001), respectively, vs. the ND group).

We next measured the mRNA expression of the fibrosis-related genes *Col1a1*, *Tgf-β1*, and *Timp1* (Figure 6C). The expression of all three was higher in the ND group than in the normal group (*Col1a1*, *p* < 0.001; *Tgf-β1*, *p* < 0.05; *Timp1*, *p* > 0.05). However, MFM and HY7207 significantly reduced the expression of *Col1a1* and *Tgf-β1* vs. the ND group. Treatment with MFM or HY7207 also tended to reduce *Timp1* expression, but there was no significant difference vs. the ND group. Thus, HY7207 reduces the expression of genes related to lipogenesis, inflammation, and fibrosis in NAFLD-inducing diet-fed mice.

In addition, we assessed the effects of HY7207 on the mRNA expression of apoptosis-related genes in the liver (Figure 6D) and found that mRNA expression of *Bax* was 1.21-fold higher in ND mice, but was reduced 0.96-fold and 0.77-fold by treatment with MFM (*p* > 0.05) or HY7207 (*p* < 0.001). The mRNA expression of *Bcl-2* in the ND group (0.81-fold) tended to be lower than that of the CON group. Treatment with MFM or HY7207 led to a significant recovery in *Bcl-2* expression, to 1.34-fold (*p* < 0.01) and 1.33-fold (*p* < 0.01), respectively, vs. the ND group. Furthermore, in NAFLD-inducing diet-fed mice, the *Bax*/*Bcl-2* ratio was significantly higher (1.53-fold, *p* < 0.01) than that of the control group. By contrast, the *Bax*/*Bcl-2* ratio of the MFM and HY7207 groups was significantly lower (0.77-fold and 0.60-fold (*p* < 0.001), respectively) than that of the ND group. Taken together, these results show that HY7207 limits apoptosis in the livers of mice fed an ND.

### 2.9. Effects of HY7207 on the Histology of Epididymal Fat

To analyze the effect of HY7207 on adipose tissue histology, we collected epididymal fat samples from the mice and examined sections of tissue after H&E staining. As shown in Figure 7A, the adipocytes of the ND group were larger than those of the control group. By contrast, the MFM and HY7207 groups had smaller adipocytes than the ND group.

### 2.10. Effects of HY7207 on the Expression of Lipogenesis-Related Genes in the Epididymal Fat of Mice with NAFLD

Finally, we determined the effects of HY7207 on genes related to lipid synthesis in epididymal fat (Figure 7B–E). The mRNA expression of *Srebp-1c*, *C*/*ebpα*, *Pparγ*, *Acaca*, and *Lipe* was significantly increased by the ND (*Srebp-1c*, *p* < 0.01; *C*/*ebpα*, *Pparγ*, *Lipe*, *p* < 0.001; *Acaca*, *p* < 0.05). The expression of *Srebp-1c* tended to be reduced by MFM and HY7207 treatment, but there were no significant differences. The mRNA expression of *C*/*ebpα*, *Acaca*, and *Lipe* in the MFM and HY7207 groups was lower than that in the ND group, but only significantly so in the case of HY7207 (*C*/*ebpα* and *Lipe*, *p* < 0.01; *Acaca*, *p* < 0.05). The expression of *Pparγ* in the MFM group was significantly lower than that of the ND group (*p* < 0.01). HY7207 treatment also significantly reduced the expression of this gene vs. the ND group (*p* < 0.001). Thus, HY7207 reduces the expression of lipid synthesis-related genes in epididymal fat, as well as in the livers, of NAFLD-inducing diet-fed mice.

## 3. Discussion

NAFLD is a common liver disease, characterized by hepatic steatosis, liver inflammation, and fibrosis [1,3]. The prevalence of NAFLD is estimated to be 20–30% worldwide, but there are no U.S. Food and Drug Administration or European Medicines Agency-approved therapies [33,34]. However, several recent studies have shown that probiotics may prevent and/or ameliorate NAFLD [30,35,36].

*L. paracasei* HY7207, isolated from a traditional Korean beverage, was selected from among 15 LAB strains because it caused significant inhibition of lipid accumulation in PA-treated HepG2 cells. We next tested the potential of HY7207 for industrial use through culture in a large-scale fermenter. In general, the bacterial growth curve is divided into four phases: the lag, exponential, stationary, and death phases [37]. The HY7207 strain growth curve could be divided into these phases as follows: 0–6 h, lag phase; 6–18 h, exponential phase; and 18–21 h, stationary phase. The growth curve of the HY7207 cells showed a typical growth curve as far as the stationary phase in industrial-scale culture. In addition, we performed whole-genome sequencing to determine the probiotic potential of HY7207, and we analyzed genetic data related to metabolic activity, antibiotic resistance, and virulence factors. The complete genome of *L. paracasei* HY7207 was 2.88 Mbp in length, with 46.43% GC content, 2778 CDSs, 15 rRNA-encoding genes, and 58 tRNA-encoding genes. In addition, no virulence factors or antibiotic resistance genes were predicted in the DNA sequence of HY7207. Taken together, these results suggest that HY7207 is a potentially useful strain that can be mass-produced.

Excessive DNL in hepatocytes is known to be a major cause of NAFLD [8]. SREBP-1c is a transcription factor that regulates the expression of genes encoding proteins in the DNL pathway and thereby increases fatty acid synthesis by FASN from acetyl-CoA and malonyl-CoA [11,38]. Previous studies have shown that hepatic DNL is upregulated in patients with NAFLD vs. healthy people, alongside higher hepatic expression of FASN [39,40]. In addition, C/EBPα plays an important role in the regulation of adipogenesis, and high expression of this protein is also associated with NAFLD [41,42]. We determined the effect of HY7207 on the expression of lipogenesis-related genes in PA-treated HepG2 cells, finding that this strain significantly increases the expression of *SREBP-1c*, *FASN*, and *C*/*EBPα* mRNA. Furthermore, hepatocyte apoptosis has been demonstrated to be a common feature of patients with NAFLD. An imbalance in BAX (a pro-apoptotic protein) and BCL-2 (an anti-apoptotic protein) expression is known to be involved in hepatocyte apoptosis during NAFLD [18,43]. Here, we found that HY7207 alters the expression of these apoptosis-related genes and the BAX/BCL-2 ratio in PA-treated HepG2 cells. These data suggest that HY7207 reduces hepatic fatty acid synthesis and apoptosis in hepatocytes.

Next, we showed that HY7207 improves physiologic and serum biochemical parameters, hepatic histology, and the gene expression in the liver and epididymal fat tissues of mice fed an ND, using MFM as a positive control. MFM is widely used for the treatment of diabetes and is known to reduce fat accumulation and ameliorate NAFLD [44]. During the experimental period, the ND induced significant increases in body mass, FER, and the masses of the liver and epididymal fat in the mice. However, treatment with HY7207 reduced these effects, and in the case of the liver and epididymal fat/body mass ratios, these were reduced to levels similar to those of the MFM group.

The circulating AST and ALT activities are markers of liver injury, are high in NAFLD/NASH, and indicate the progression of the disease [45]. The serum ALP activity may also be a predictor of significant fibrosis in patients with obesity and NAFLD [46]. In addition, γ-GPT is an indicator of liver injury, and a decrease in γ-GPT activity is associated with an improvement in liver histology [47]. The serum activities of AST, ALT, ALP, and γ-GPT were high in mice fed the ND, whereas MFM and HY7207 significantly reduced these. This is further evidence that treatment with HY7207 ameliorates NAFLD. High serum TG, T-CHOL, and LDL-C concentrations are associated with NAFLD, and these defects were present in the ND group [48,49]. However, both MFM and HY7207 reduced their concentrations. NAFLD is associated with excessive hepatic glucose production, resulting in hyperglycemia, and therefore the circulating glucose concentration is also a potential marker of the development of NAFLD [50]. This was high in the ND group, but slightly lower in the HY7207 group. However, the serum HDL-C concentration did not show significant differences between the groups. In addition, analysis of the hepatic histology of the mice showed that HY7207 reduces the steatosis and NAS score (steatosis + lobular inflammation + hepatocyte ballooning) of mice with NAFLD [51]. These results imply that HY7207 improves the serum lipid profile and ameliorates the hepatic steatosis of mice, suggesting that this probiotic reduces liver fat accumulation and may alleviate the symptoms of NAFLD.

Hepatic fat accumulation induces the development of inflammation and leads to the production of pro-inflammatory cytokines and chemokines [19]. This liver inflammation is an important precursor to liver fibrosis in NAFLD [52]. The hepatic mRNA expression of the lipogenic genes *Srebp1c*, *Fasn*, *Cebpa*, *Pparg*, and *Acaca* of the mice was lower in the HY7207 group than in the ND group. In addition, there was high expression of genes encoding pro-inflammatory cytokines (*Tnfa* and *Il1b*) and the chemokine CCL2 in the ND group, while treatment with HY7207 significantly reduced their expression. Furthermore, the expression of the fibrosis-related genes *Col1a1*, *Tgfb1*, and *Timp1* was lower in the HY7207 group than in the ND group These data imply that HY7207 ameliorates hepatic lipid accumulation, steatosis, inflammation, and fibrosis in mice with NAFLD. Hepatic apoptosis is considered to be a major cause of the liver inflammation and fibrosis. Similar to the in vitro experiments, HY7207 treatment had positive effects on the expression of the apoptosis-related genes *Bax* and *Bcl2* and the *Bax*/*Bcl2* ratio in the animal model. Histopathologically, hepatocyte ballooning, a form of hepatocyte death, is generally considered to indicate apoptosis [53,54]. In the present study, the effects of HY7207 on the expression of apoptosis-related genes were consistent with its effects on hepatocyte ballooning on H&E-stained sections.

NAFLD is closely associated with excessive calorie intake in the form of carbohydrates or fats and obesity [55]. The long-term excessive intake of either can lead to hyperinsulinemia and hyperglycemia. This is associated with greater breakdown of triglyceride in adipose tissue, which increases the amount of free fatty acids entering the liver [56,57,58]. Therefore, a better understanding of the liver–adipose tissue crosstalk in NAFLD may be necessary for the development of improved approaches to the treatment and prevention of NAFLD. In the present study, the hypertrophy of adipocytes and lipid synthesis related gene expression in the epididymal fat in the ND group was reduced by HY7207 administration (Figure 7). This may suggest that HY7207 may also ameliorate NAFLD through a distinct mechanism. However, the molecular mechanism whereby HY7027 affects crosstalk between the liver and adipose tissue and thereby influences NAFLD should be the subject of further in-depth studies.

## 4. Materials and Methods

### 4.1. Sample Collection and Isolation of LAB Strains

LAB strains were obtained from Makgeolli, a traditional Korean fermented beverage. Samples were purchased from traditional Korean markets, collected in sterile containers, and stored in a −80 °C freezer until used. For strain isolation, liquid samples were mixed with sterile phosphate-buffered saline (PBS) in a ratio of 1:9 (*v*/*v*). The mixed samples were serially diluted with sterile PBS at 1/10 ratio, spread on MRS agar plates (BD Difco, Sparks, MD, USA), and incubated under anaerobic conditions at 37 °C for 48 h. Fifteen colonies were randomly selected and streaked onto fresh MRS agar plates to obtain pure isolates. These strains were then stored at −80 °C in glycerol (20%, *v*/*v*).

### 4.2. Culture of HepG2 Cells

HepG2 human liver cancer cells were purchased from the American Type Culture Collection (Manassas, VA, USA). The cells were cultured at 37 °C in a 5% CO_2_ incubator in Minimum Essential Medium (MEM; Welgene, Gyeongsan, Republic of Korea) containing 10% fetal bovine serum (Gibco, Waltham, MA, USA) and 1% penicillin/streptomycin (Gibco, Waltham, MA, USA). The HepG2 cells used in the experiment had undergone ≤10 passages.

### 4.3. Assessment of the Inhibition of Lipid Accumulation in HepG2 Cells

To evaluate the abilities of the 15 selected strains to inhibit lipid accumulation, Oil-red-O (ORO) staining was performed on PA-treated HepG2 cells. HepG2 cells were seeded onto 12-well plates at a density of 1 × 10^5^ cells/well and acclimated for 24 h, then 0.75 mM PA-containing antibiotic-free MEM was added to all the wells except the control wells, and each LAB strain (10^7^ CFU/mL) was added. The control wells were treated with antibiotic-free MEM containing saline, and negative controls were treated with 0.75 mM PA alone. After 24 h, each well was washed twice with PBS, and 1 mL of 10% formalin was added, followed by incubation for 1 h. The wells were then rinsed with distilled water, and the cells were treated with 60% isopropanol and incubated for 5 min. After removing the isopropanol, 400–500 µL of 60% ORO solution was added, and the cells were incubated for 10 min. After that, the wells were washed with tap water and dried, then 100% isopropanol was added, and the cells were further incubated for 5 min. The isopropanol was then transferred to a microcentrifuge tube and centrifuged at 10,000× *g* for 2 min. The absorbance of the supernatant was measured at 480 nm using a microplate reader.

### 4.4. Industrial Culture and Assessment of the Growth Curve of HY7207 Cells

HY7207 cells were cultured in a 2000-L plant-scale fermenter (CNS, Daejeon, Republic of Korea) to assess the possibility of industrial scale-up and to confirm appropriate bacterial growth. A 1200-L culture was prepared at 37 °C and pH 5.5, with rotation at 100 rpm, for 21 h. The culture medium was sampled every 1 h, and the number of bacteria present was measured by two methods. First, the culture medium was serially diluted in saline for a standard plate count to be performed. The dilutions were incubated with MRS in petri dishes for 48 h, then 30–300 colonies were checked and expressed as CFU/mL.

### 4.5. Whole Genome Sequencing of HY7207

Genomic DNA (gDNA) of HY7207 was isolated by the Qiagen MagAttract HMW DNA Kit (Qiagen, Hilden, Germany), and quality and quantity of gDNA was measured by the Qubit 2.0 fluorometer (Invitrogen, Carlsbad, CA, USA). Sequencing of purified gDNA was performed at CJ bioscience (Seoul, Republic of Korea) using Illumina Miseq (Illumina Inc., San Diego, CA, USA) and PacBio RS Ⅱ (Pacific Biosciences Inc., Menlo Park, CA, USA) sequencing platforms. DNA libraries were prepared by the TruSeq DNA Library LT Kit (Illumina Inc., San Diego, CA, USA) and the SMRTbell Template Preparation Kit (Pacific Biosciences Inc., Menlo Park, CA, USA). Gene prediction was performed by Prodigal 2.6.2, and Cluster of Orthologous (COG) of predicted gene was annotated by EggNOG 4.5 (http://eggnogdb.embl.de (accessed on 8 June 2024)). Antibiotic resistance genes were predicted using The Comprehensive Antibiotics Resistance Database (CARD; https://card.mcmaster.ca/ (accessed on 8 June 2024)).

### 4.6. Animal Experiments

Six-week-old male C57BL/6 mice were purchased from Dooyeol Biotech (Gyeonggi, Republic of Korea) and acclimated for 1 week. The mice were housed under constant humidity and temperature conditions (55% ± 10%, 22 ± 1 °C) under a 12 h light/dark cycle. After acclimation, the mice were randomly assigned to one of four groups, each containing seven mice: control (CON, AIN-93G diet), NAFLD diet only (ND), NAFLD diet with metformin (MFM, 250 mg/kg/day), and NAFLD diet with HY7207 (HY7207, 10^9^ CFU/kg/day). The ND comprised a rodent diet providing 40 kcal% as fat (principally palm oil), 20 kcal% as fructose, and 2% as cholesterol (D09100310, Research Diets, Inc., New Brunswick, NJ, USA). The MFM and probiotic were dissolved in 200 uL of saline and orally administered for 8 weeks, and the Control and ND groups were administered an equal volume of saline over the same period. The body masses and food intakes of the mice were measured weekly. After 8 weeks, the mice were sacrificed, and blood, liver, and epididymal fat tissue samples were obtained. The blood samples were allowed to stand at room temperature and then centrifuged at 3000× *g* for 20 min to separate the serum. The separated serum samples were then used for biochemical analysis. The other tissue samples were rinsed with PBS, weighed, and stored at −80 °C until subsequent analysis. The animal studies were approved by the Institutional Animal Care and Committee of the hy Co., Ltd. R&D Center, Seoul, Republic of Korea (approval number: AEC-2024-00001-Y). Figure 8 shows a flowchart of the animal experiments performed in the present study.

### 4.7. Blood Biochemical Analysis

The serum activities/concentrations of AST, ALT, ALP, γ-GTP, T-CHOL, TG, GLU, HDL, and LDL were analyzed at T&P Bio (Gyeonggi, Republic of Korea).

### 4.8. Histological Analysis

Liver and epididymal fat samples were collected and fixed in 10% (*v*/*v*) formalin solution. The fixed tissues were embedded in paraffin, and sections were cut and mounted on slides. Images were then obtained using a Zeiss Axiovert 200M microscope (Carl Zeiss AG, Thornwood, NY, USA) after H&E staining. The NAS of the liver samples, comprising hepatic steatosis, lobular inflammation, and hepatocyte ballooning, was evaluated [51]. The degree of hepatic steatosis was graded 1–4 as follows: 1, <5%; 2, 5–33%; 3, 34–65%; and 4, >66%. The degree of lobular inflammation was graded 0–3 as follows: 0, no foci; 1, <2 foci/×200 field; 2, 2–4 foci/×200 field; and 3, >4 foci/×200 field. The degree of hepatocyte ballooning was graded 0–2 as follows: 0, none; 1, few ballooned cells; 2, many ballooned cells/prominent ballooning. Histopathologic analysis, including H&E staining and NAS scoring, were performed by DooYeol Biotech.
NAS = Hepatic steatosis + Lobular inflammation + Hepatocyte ballooning

### 4.9. Isolation of RNA, cDNA Synthesis, and Real-Time PCR

RNA was extracted from HepG2 cells and mouse livers using an Easy-spin Total RNA Extraction Kit (iNtRON Biotechnology, Seoul, Republic of Korea), then cDNA was synthesized from 2 μg RNA using an Omniscript RT Kit (Qiagen, Hilden, Germany) at 37 °C for 60 min. The cDNA template was analyzed using the QuantStudio 6 Flex Real-time PCR System (Applied Biosystems, Foster City, CA, USA), and real-time PCR was performed using a TaqMan™ Gene Expression Master Mix (Applied Biosystems, Waltham, MA, USA). Table 2 lists the target genes and assay catalog numbers. The expression levels of the target genes were normalized to that of GAPDH (HepG2 cells, Hs99999905_m1; Mice, Mm99999915_g1) using the comparative Cτ method.

### 4.10. Statistical Analysis

Data are presented as the mean ± standard deviation (SD). Groups were compared using one-way ANOVA, followed by Tukey’s post hoc test. All analyses were conducted using Prism v.6.0 (GraphPad Software, San Diego, CA, USA), and *p* < 0.05 was considered to represent statistical significance.

## 5. Conclusions

*Lacticaseibacillus paracasei* HY7207, isolated from a traditional Korean fermented beverage, can be cultured at an industrial scale. The data presented herein suggest that HY7207 has potential for use as a functional food supplement for the amelioration of NAFLD. HY7207 ameliorates hepatic fat accumulation in PA-treated HepG2 cells and mice with NAFLD. In addition, HY7207 may ameliorate the symptoms of NAFLD by reducing the expression of lipogenesis and apoptosis-related genes. Furthermore, HY7207 has anti-inflammatory and anti-fibrotic effects by reducing the expression of genes encoding cytokines, chemokines, and mediators of liver fibrosis. Therefore, we propose that HY7207 could be produced by the food industry as a functional supplement for patients with NAFLD.

## Figures and Tables

**Figure 1 ijms-25-09870-f001:**
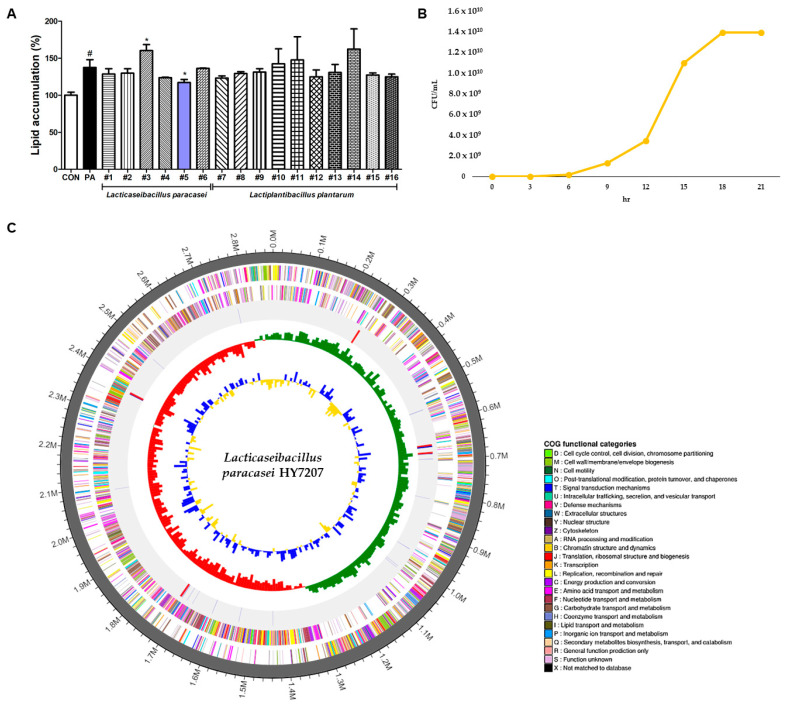
(**A**) Screening of 15 LAB strains for their ability to reduce lipid accumulation in PA-treated HepG2 cells. (**B**) Growth curve of HY7207 in industrial-scale culture. (**C**) Genome map of *Lactocaseibacilllus paracasei* HYY7207. From the outer circle to the inner circle, the genomic features are indicated as follows: genome size, gray color; forward strand and reverse strand CDSs, different colors, according to the COG classification; tRNA, blue line; rRNA, red line; GC skew, green and red peaks; GC ratio, blue and yellow peaks. Data are presented as the mean ± SD. ^#^ *p* < 0.05 vs. the control group, * *p* < 0.05 vs. the PA group; LAB: lactic acid bacteria; PA: palmitic acid; CDS: protein-coding DNA sequence; COG: clusters of orthologous groups; G: guanine, C: cytosine.

**Figure 2 ijms-25-09870-f002:**
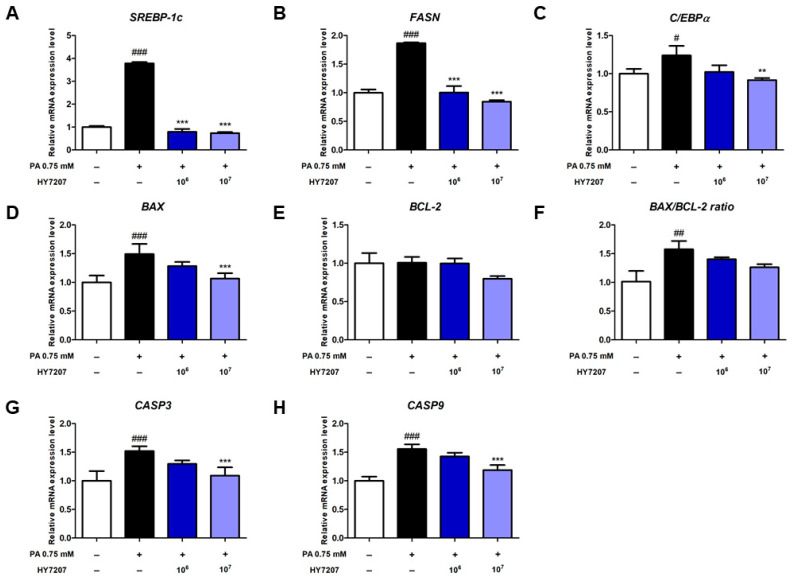
Effect of HY7207 on the expression of lipogenesis and apoptosis-related genes in PA-treated HepG2 cells. (**A**) *SREBP-1c*, (**B**) *FASN*, (**C**) *C*/*EBPα*, (**D**) *BAX*, (**E**) *BCL-2*, (**F**) *BAX*/*BCL-2* ratio (**G**) *CASP3*, and (**H**) *CASP9*. The results are presented as the mean ± SD. ^#^ *p* < 0.05, ^##^ *p* < 0.01, and ^###^ *p* < 0.001 vs. the control group; ** *p* < 0.01 and *** *p* < 0.001 vs. the PA group. PA: palmitic acid; HY7207: *Lacticaseibacillus paracasei* HY7207; SREBP-1c: sterol regulatory element-binding protein 1; FASN: fatty acid synthase; C/EBPα: CCAAT/enhancer-binding protein alpha; BAX: BCL2-associated X, apoptosis regulator; BCL-2: BCL2, apoptosis regulator; CASP3: Caspase 3; CASP9: Caspase 9.

**Figure 3 ijms-25-09870-f003:**
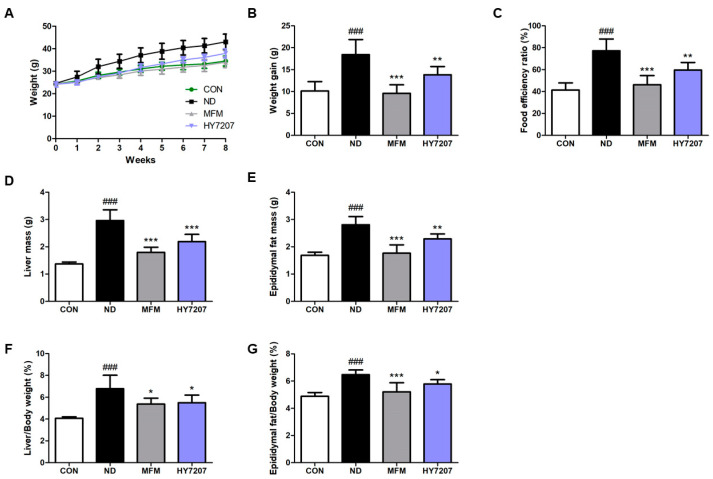
Effect of HY7207 on mice fed an NAFLD-inducing diet. (**A**) Body mass changes, (**B**) body mass gain, (**C**) food efficiency ratio, (**D**) liver tissue mass, (**E**) liver/body mass ratio, (**F**) epididymal fat mass, and (**G**) epididymal fat/body mass ratio. The results are presented as the mean ± SD. ^###^ *p* < 0.001 vs. the CON group; * *p* < 0.05, ** *p* < 0.01, and *** *p* < 0.001 vs. the ND group. CON: Untreated group; ND: NAFLD-inducing diet group; MFM: metformin; HY7207: *Lacticaseibacillus paracasei* HY7207; FER: food efficiency ratio.

**Figure 4 ijms-25-09870-f004:**
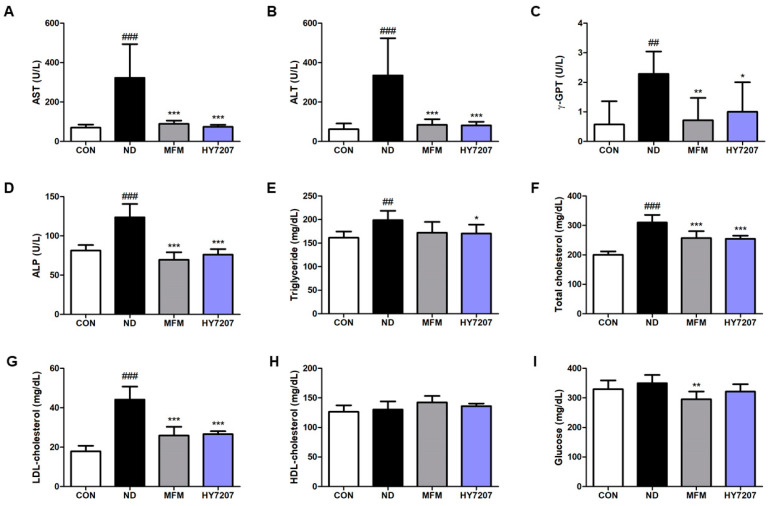
Effect of HY7207 on the blood biochemistry of NAFLD-inducing diet-fed mice. (**A**) AST, (**B**) ALT, (**C**) ALP, and (**D**) γ-GPT activities; (**E**) TG, (**F**) T-CHO, (**G**) LDL-C, (**H**) HDL-C, and (**I**) GLU concentrations. The results are presented as the mean ± SD. ^##^ *p* < 0.01 and ^###^ *p* < 0.001 vs. the CON group; * *p* < 0.05, ** *p* < 0.01, and *** *p* < 0.001 vs. the ND group. CON: Untreated group; ND: NAFLD-inducing diet group; MFM: Metformin group; HY7207: *Lacticaseibacillus paracasei* HY7207 group; AST: aspartate aminotransferase; ALT: alanine aminotransferase; ALP: alkaline phosphatase; γ-GTP: gamma-glutamyl transferase; TG: triglyceride; T-CHO: total cholesterol; LDL-C: low-density lipoprotein-cholesterol; HDL-C: high-density lipoprotein-cholesterol; GLU: glucose.

**Figure 5 ijms-25-09870-f005:**
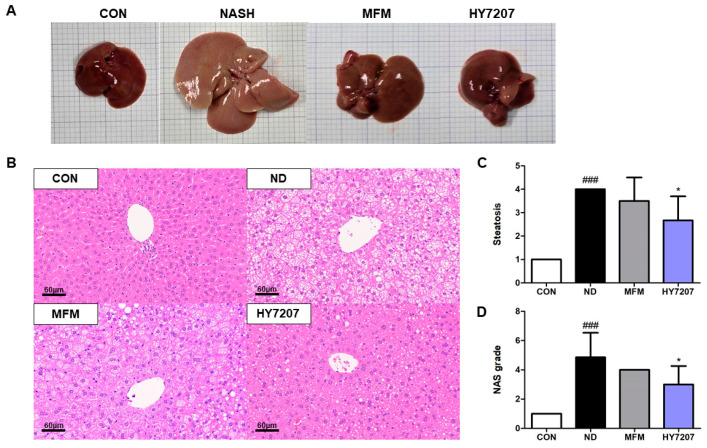
Effect of HY7207 on the hepatic histology of NAFLD-inducing diet-fed mice. (**A**) Liver morphology, (**B**) histology of the liver (hematoxylin and eosin-stained sections; 100× magnification), and (**C**) steatosis grade of the mice. (**D**) NAS grade of the mice. Results are presented as the mean ± SD. ^###^ *p* < 0.001 vs. the CON group; * *p* < 0.05 vs. the ND group. CON: Untreated group; ND: NAFLD-inducing diet-fed group; MFM: Metformin group; HY7207: *Lacticaseibacillus paracasei* HY7207 group; NAS: NAFLD activity score.

**Figure 6 ijms-25-09870-f006:**
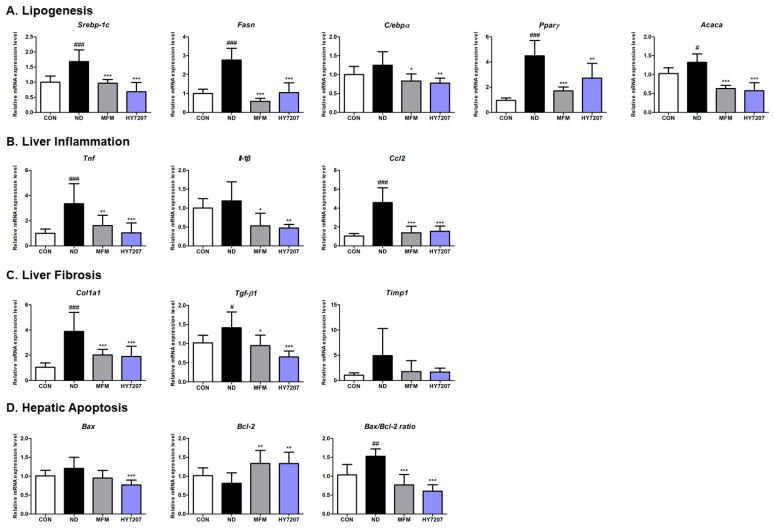
Effect of HY7207 on the hepatic gene expression of NAFLD-inducing diet-fed mice. The expression of genes related to (**A**) lipogenesis, (**B**) inflammation, (**C**) fibrosis, and (**D**) hepatic apoptosis is shown. Results are presented as the mean ± SD. ^#^ *p* < 0.05, ^##^ *p* < 0.01, and ^###^ *p* < 0.001 vs. the CON group; * *p* < 0.05, ** *p* < 0.01, and *** *p* < 0.001 vs. the ND group. CON: Untreated group; ND: NAFLD-inducing diet-fed group; MFM: Metformin group; HY7207: *Lacticaseibacillus paracasei* HY7207 group.

**Figure 7 ijms-25-09870-f007:**
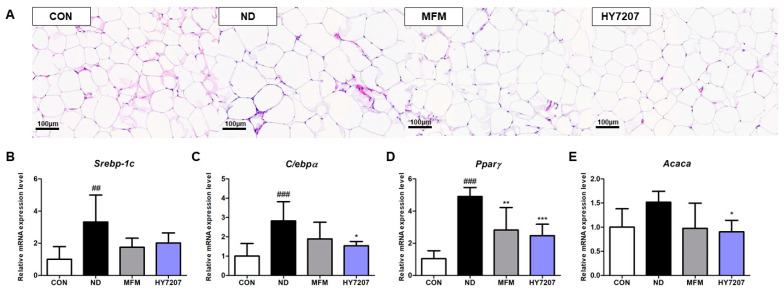
(**A**) Effects of HY7207 on the histology of the epididymal fat of NAFLD-inducing diet-fed mice. (H&E staining; 100× magnification). Effect of HY7207 on the expression of lipogenesis related genes in NAFLD-inducing diet-fed mice. (**B**) *Srebp-1c*, (**C**) *C*/*ebpα*, (**D**) *Pparγ* and (**E**) *Acaca*. Results are presented as the mean ± SD. ^##^ *p* < 0.01, and ^###^ *p* < 0.001 vs. the CON group; * *p* < 0.05, ** *p* < 0.01, and *** *p* < 0.001 vs. the ND group. CON: Untreated group; ND: NAFLD-inducing diet-fed group; MFM: Metformin group; HY7207: *Lacticaseibacillus paracasei* HY7207 group.

**Figure 8 ijms-25-09870-f008:**
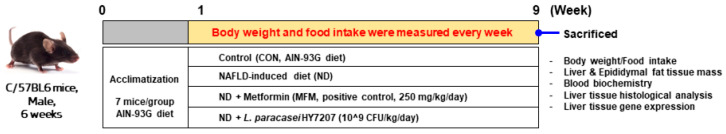
A flow chart of animal experiments.

**Table 1 ijms-25-09870-t001:** Genomic information of *Lacticaseibacillus paracasei* HY7207.

Features	Terms
Sequencing platforms	Illumina MiSeq PacBio RSII
Libraries used	TruSeq DNA Library LT Kit SMRTbell^®^ Prep Kit
Genome size (bp)	2,877,365
G+C contents (%)	46.43
rRNA genes	15
tRNA genes	58

**Table 2 ijms-25-09870-t002:** Lists of TaqMan probes used for gene expression analysis.

Gene	Gene Name	Catalog Number
In vitro		
*GAPDH*	Glyceraldehyde-3-phosphate dehydrogenase	Hs99999905_m1
*SREBP-1c*	Sterol regulatory element-binding protein 1	Hs01088691_m1
*FASN*	Fatty acid synthase	Hs00188012_m1
*C*/*EBPα*	CCAAT/enhancer-binding protein alpha	Hs00269972_s1
*BAX*	BCL2 associated X, apoptosis regulator	Hs00180269_m1
*Bcl-2*	BCL2, apoptosis regulator	Hs04986394_s1
*CASP3*	Caspase 3	Hs00234387_m1
*CASP9*	Caspase 9	Hs00962278_m1
In vivo		
*Gapdh*	Glyceraldehyde-3-phosphate dehydrogenase	Mm99999915_g1
*Srebp-1c*	Sterol regulatory element-binding protein 1	Mm00550338_m1
*Fasn*	Fatty acid synthase	Mm00433237_m1
*C*/*ebpα*	CCAAT/enhancer-binding protein alpha	Mm00514283_m1
*Pparγ*	Peroxisome proliferator-activated receptor gamma	Mm00440945_m1
*Acaca*	Acetyl-CoA carboxylase alpha	Mm01304257_m1
*Tnf*	Tumor necrosis factor	Mm00443258_m1
*Il-1β*	Interleukin 1 beta	Mm00434228_m1
*Ccl2*	C-C motif chemokine ligand 2	Mm00441242_m1
*Col1a1*	Collagen type I alpha 1	Mm00801666_g1
*Tgf- β1*	Transforming growth factor beta 1	Mm01178820_m1
*Timp1*	tissue inhibitor of metalloproteinase 1	Mm01341361_m1
*Bax*	BCL2 associated X, apoptosis regulator	Mm00432051_m1
*Bcl-2*	BCL2, apoptosis regulator	Mm00477631_m1

## Data Availability

The data presented in this study are available in the article and Appendix A.

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
