# Peer review of "Lacticaseibacillus paracsei HY7207 Alleviates Hepatic Steatosis, Inflammation, and Liver Fibrosis in Mice with Non-Alcoholic Fatty Liver Disease"

_ijms, 2024, doi:10.3390/ijms25189870_

Round 1

Reviewer 1 Report

Comments and Suggestions for Authors

The paper is well-written, and the authors analyzed various indicators of NAFLD improvement in mice following oral administration of HY7207 through animal experiments. However, there are some concerns that need clarification.

Specific points:

1.        Please explain the rationale for using hepatic steatosis in liver cell assays (in vitro) as a screening platform for LAB strains. Although LAB strains selected through this method have demonstrated the ability to improve NAFLD in animal models, additional justification for this approach is needed. Please clarify if probiotics have the potential to directly affect hepatocytes in vivo. Most importantly, Probiotics generally exert their effects through mechanisms such as modulation of the gut microbiota…

2.        In the in vitro liver cell model, the authors demonstrated that HY7207 reduced the expression of genes related to lipid synthesis and hepatocyte apoptosis in palmitic acid-treated HepG2 cells. Additionally, in animal models, HY7207 was shown to decrease the expression of genes associated with hepatic lipid synthesis (Srebp1c, Fasn, C/ebpa, Pparg, and Acaca), inflammation (Tnf, Il1b, and Ccl2), and fibrosis (Col1a1, Tgfb1, and Timp1). However, is there a possibility that HY7207 directly affects hepatocytes in vivo? It is recommended to provide more discussion on the precise mechanisms through which HY7207 improves NAFLD, including its effects on gut permeability, insulin tolerance tests (ITT), glucose tolerance tests (GTT), and inflammation in intestinal cells.

3.        In Figure 3, the authors demonstrated that the food efficiency ratio (FER) in the ND group was significantly higher than in the control group, whereas the FER in the positive control and HY7207 treatment groups was significantly lower than in the ND group. The authors also confirmed in Supplementary S1 that there were no statistically significant differences in food intake among the different groups. What is the primary mechanism by which HY7207 reduces food efficiency? Please provide a more detailed explanation.

4.        In the animal experiments, the authors mentioned, “The MFM and probiotic were dissolved in saline and orally administered for 8 weeks, and the Control and ND groups were administered saline.” Was the oral administration conducted via drinking water or oral gavage? Please provide additional details. How did the authors ensure that the animals consumed an adequate and equal amount of probiotics?

5.        This study did not analyze gut microbiota. Are the observed improvements in the animal experiments related to changes in the gut microbiota? It is suggested to analyze gut microbiota to clarify the mechanism of action of HY7207.

6.        Has the effect of HY7207 on colon inflammation markers been analyzed? Since this study involved oral administration of probiotics to NAFLD mice and only assessed the impact on hepatocytes, this is insufficient to explain the mechanism of action of HY7207. Please clarify.

Author Response

Dear. Reviewer

Thank you for your kindly advice. We have written your comment and a response to it. And we attached a PDF file about it. Please kindly confirm our responses and answers through attached PDF file.

Sincerely, Ms. Kim

Reviewer 2 Report

Comments and Suggestions for Authors

In this manuscript the authors evaluated the effect of HY7207 (a probiotic, bacterial strain) on Nonalcoholic fatty liver disease. The authors showed that HY7207 had an overall positive impact on fatty liver and ameliorated various markers of NAFLD.

The paper is well thought out and the results are convincing.

Here are my specific comments:

1.       The introduction is good. However, it would be good to know what was the rationale behind using the particular Korean fermented drink? Has it been shown to have beneficial effects on NAFLD? Why was the decision to extract probiotics from this particular drink was made?

2.       I would recommend the authors to mention the concentration of PA, metformin or HY7207, upfront while writing the results section. Although most of these have been discussed in the materials and methods section, it is the last section of the manuscript. It would be helpful to the readers to know about them beforehand. The authors have mentioned the concentration in the results section for some of the figures but, not for the others.

3.       What is the benefit of one method over the other? Or why were both the methods used for result section 2.2?

4.       The authors throughout the paper have used 3 different CFU levels for HY7207. What are the relevant levels of these microbes?

5.       Figure 2 (section 2.4) I would also recommend looking into caspase expression.

6.       Figure 3 (section 2.5) – A comment on the food intake pattern by the animals would be helpful. 

7.       Section 2.6 (lines 179), I believe these are only the concentrations of the various biochemistries and not the activities being tested. Please clarify.

8.       Figure 5 C and D - Some error bars are missing.

Comments on the Quality of English Language

N/A

Author Response

(The authors gave the same response as above.)

Round 2

Reviewer 1 Report

Comments and Suggestions for Authors

I find the author's response to be highly satisfactory.